# Effect of Different Cover Crops on Suppression of the Weed *Oxalis pes-caprae* L., Soil Nutrient Availability, and the Performance of Table Olive Trees 'Kalamon' cv. in Crete, Greece

Nikolaos Volakakis [1,2,3,*], Emmanouil M. Kabourakis [1,*], Leonidas Rempelos [3,4], Apostolos Kiritsakis [5,6] and Carlo Leifert [7,8]

1   Olive and Agroecological Production Systems Lab (EOPS), Hellenic Mediterranean University, P.O. Box 1939 Estavromenos, GR 71410 Iraklio, Greece
2   Geokomi plc, P.O. Box 21, Sivas Festos, GR 70200 Crete, Greece
3   Nafferton Ecological Farming Group, Newcastle University, Newcastle upon Tyne NE1 7RU, Tyne and Wear, UK
4   Lincoln Institute for Agri-Food Technology, University of Lincoln, Riseholme Park, Lincoln LN2 2LG, Lincolnshire, UK
5   International Hellenic University, Thermi, 57001 Thessaloniki, Greece
6   International Observatory of Oxidative Stress for Health and Agricultural Products, 21 Kar. Diehl Str, 54623 Thessaloniki, Greece
7   SCU Plant Science, Southern Cross University, Military Rd., Lismore, NSW 2480, Australia
8   Department of Nutrition, IMB, University of Oslo, 0372 Oslo, Norway
*   Correspondence: nvolakakis@gmail.com (N.V.); ekabourakis@hmu.gr (E.M.K.)

**Abstract:** Winter cover crops are used in organic olive production to increase N-supply and yields, and to reduce weed competition. However, there is limited information on the effect of different cover crops on weed suppression, soil fertility and productivity of organic olive orchards. Here, we compared the relative effect of four contrasting cover crops established from (i) untreated vetch seed, (ii) vetch seed inoculated with a commercial *Rhizobium* seed inoculum, (iii) an untreated vetch/barley/pea seed mixture and (iv) untreated seed of *Medicago polymorpha* L. (a native legume species which establishes naturally in olive orchards in Crete) in a 35-year-old experimental table olive orchard. The use of a vetch/barley/pea mixture resulted in the greatest suppression of the dominant weed species *Oxalis pes-caprae*. *Rhizobium* inoculation of vetch seed resulted in significantly lower vetch establishment and significantly higher *Oxalis* suppression but had no significant effect on the root nodulation of vetch plants. There was no significant difference in fruit yield between cover crop treatments, but the fruit weight was significantly higher when cover crops were established from un-treated vetch seeds and the vetch/barley/pea seed mixture compared with the cover crops based on inoculated vetch or Medicago seed. However, although *Medicago* establishment was very low (<10 plants/m²), fruit yields were numerically 20% higher in the Medicago plots. These findings suggests that, overall, legume cover crops had no effect on fruit yields. This conclusion is supported by the results of the olive leaf analyses which detected no significant differences in nitrogen and other mineral macro- and micronutrient concentration between treatments, except for B (highest in olive leaves from Medicago and lowest in untreated vetch plots) and Mo (highest in olive leaves from Medicago and lowest in vetch/barley/pea mixture plots). Overall, our results suggest that the current recommendation to establish legume-based cover crops in organic olive orchards every year, may need to be revised and that establishing cover crops every 2–4 years may reduce costs without affecting olive fruit yields.

**Keywords:** organic olive production; ground cover crops; vetch; *Medicago*; *Oxalis*; weed suppression; olive fruit yield; leaf nutrient analysis; *Rhizobium* inoculum; nodulation

## 1. Introduction

Winter legume cover crops are used to increase nitrogen supply and suppress Oxalis (*Oxalis pes-caprae* L.), the dominant weed species found in organic olive orchards in Crete. The use of legumes is recommended, because (a) there is often limited availability of organic fertilisers such as animal manure, (b) commercial fertiliser products that are permitted under organic production standards are relatively expensive and (c) many organic fertilisers have a low plant available nitrogen content [1–4]. However, fertilisation regimes based on legume winter cover crops are thought to result in lower nitrogen availability than that achieved by standard mineral fertilisation regimes used in conventional farming [4–6].

One approach to increase N-fixation and availability from cover crops may be to apply *Rhizobium* inoculum to legume seed. For example, the application of a commercial *Rhizobium* inoculum to clover seed was recently shown to further increase N-levels in soil and N-supply to subsequent wheat crops grown after clover leys in the UK [4,7]. However, this approach has not been evaluated for vetch (*Vicia sativa*), the main legume species used as cover crop in organic olive orchards in the Mediterranean region [8].

The use of native legume species instead of vetch has also been suggested as a strategy to increase N-supply in organic olive production, since they may be better adapted to establishment, growth and nitrogen fixation under local conditions [1–3]. For example, in Crete, *Medicago polymorpha* L. is the main native legume species found in olive orchards. However, due to the common practice of incorporating cover crops in April/May to minimize wildfire risk, *Medicago* rarely establishes well in commercial olive orchards, because it produces seeds in late spring/early summer. However, *Medicago* rapidly re-established in abandoned and non-tilled olive orchards. The relative efficacy of N-fixation by *M. polymorpha* and *V. sativa*-based cover crops has not been compared [3].

Cover crop mixtures consisting of different legumes and deeper rooting cereal or grass species (which are thought to minimise N-leaching losses during the winter rain period) may also improve N-fixation and/or retention in soil but cover crop mixtures have so far not been evaluated in organic olive production systems [1,3].

The main objectives of this study were therefore to compare the effect of vetch cover crops established from untreated *V. sativa* seed (which is the standard cover crop currently used in commercial organic olive orchards) with three novel cover crops (a vetch/barley/pea mixture, *Medicago polymorpha* L. and vetch established from *Rhizobium*-inoculated seed) on the (i) establishment and plant density of the dominant weed species *Oxalis pes-caprae,* (ii) availability pattern of mineral macro- and micronutrients to olive trees (via leaf analysis immediately after cover crops were incorporated in May and in the following October during fruit development) and (iii) yield and fruit weight/size of table olives in two harvest years (olives are managed/pruned to achieve a biennial cropping pattern in the Messara region).

## 2. Materials and Methods

### 2.1. Experimental Orchard Used

The field experiment was carried out within an experimental table olive orchard at the National Agricultural Research Foundation of Greece (NAGREF). The orchard is located 8 km east (latitude 35°3′27.33″ N, longitude 24°56′18.22″ E) of the town of Moires in the Messara plain in southern Crete, Greece (Figure 1). The orchard was at 158 m O.D. and had been planted with 900 'Kalamon' cv. and 388 'Manzanila' cv. trees in 1975. At the time of the experiment the trees were 35 years old and had a height of 3.5–4 m and were planted 6 m apart. The orchard was in a landscape dominated by commercial olive fields (> 50% of agricultural land area) and areas with wild olive trees and abandoned orchards.

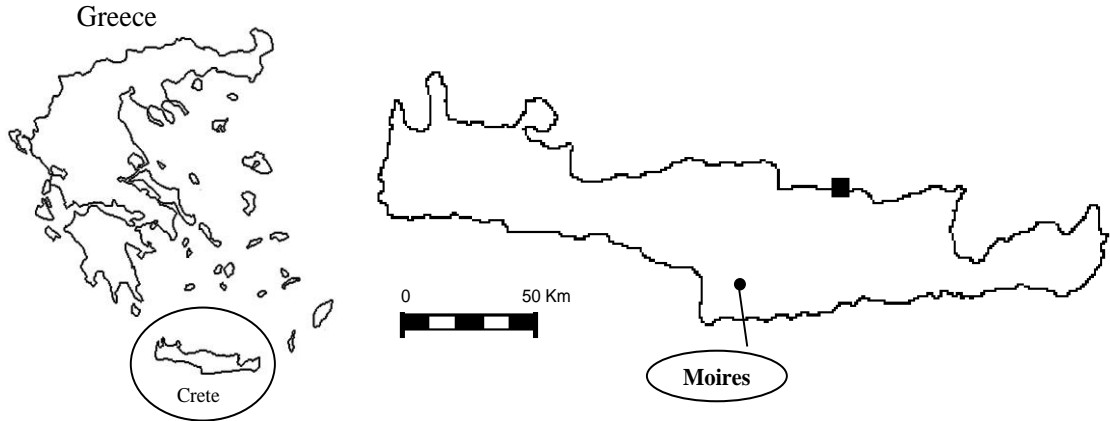

**Figure 1.** Location of experimental orchards in Crete, Greece.

*2.2. Experimental Design and Cover Crop Treatments*

A randomised block design was used incorporating four blocks of 48 'Kalamon' olive trees, each split into four treatment plots (12 trees/plot). In treatment plots four different cover crop treatments were applied for 3 consecutive growing seasons (2005/2006, 2006/2007 and 2007/2008) to compare their effect on cover crop and *Oxalis* establishment, invertebrate activity (results not reported here), olive yields, mineral supply to olive trees and olive fly infestation. The four cover crop treatments were: (i) vetch (*Vicia sativa*) without *Rhizobium* inoculation; (ii) a mixture of vetch (*Vicia sativa*), pea (*Pisum sativum*) and barley (*Hordeum vulgare*); (iii) vetch (*Vicia sativa*) with *Rhizobium* inoculation (Legume Fix, Legume Technology Ltd., Nottinghamshire, UK); and (iv) a native wild *Medicago* species (*Medicago polymorpha*). Vetch seed used in treatments i, ii and iii were of the 'Alexandros' cv., peas used in treatment ii were of the 'Dodoni' cv. and barley used in treatment ii was an unnamed local variety which had been produced from farm saved seed for many years in the Messara area. *Medicago* seeds were collected from olive orchards in the area. Seed germination rates of the cover crop species used in treatments were 99.7%, 99.3%, 99.3% and 55% for vetch, pea, barley and *Medicago* seeds, respectively. The plot dimension/size is shown in Figure 2 and climatic conditions (monthly rainfall and average mean daily temperature) during the three cover-crop and olive growing seasons are summarised in Figure 3.

*2.3. Orchard Management*

The orchard was managed commercially in accordance with EU organic farming standards [9] based on agronomic protocols used in the area since 1993 [1]. The orchard soil was shallow (to 10 cm depth) ploughed before sowing of cover crops in early December using a Universal 643 DT tractor (Universal (UTB), Brasov, Romania) and a Tiger chisel plough-type cultivator (TIGER SA, Heraklion, Greece), which does not invert the soil. This was performed to incorporate manure and spontaneous ground cover vegetation into the soil. Soils were cultivated again using the same tractor and a Pythagoras rotatory cultivator (rotavator) (Pythagoras SA, Thessaloniki, Greece) in the middle of April, to break up and incorporate the cover crop and other spontaneous vegetation into the soil. This was performed to increase soil organic matter levels and mineralisation potential, minimise nitrogen loss, maximise water infiltration and retention (thus minimising soil

water loss) and to minimise dry vegetation residues present on the soil surface in order to limit orchard damage from wildfires during the summer months [9].

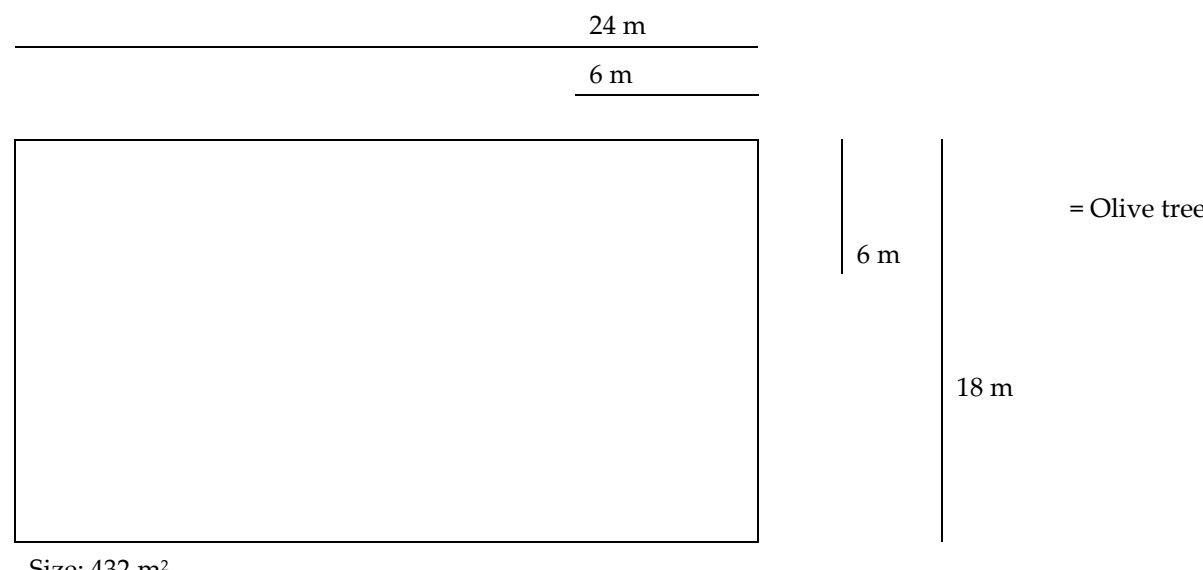

**Figure 2.** Experimental plot dimensions. , location of individual olive trees in plots.

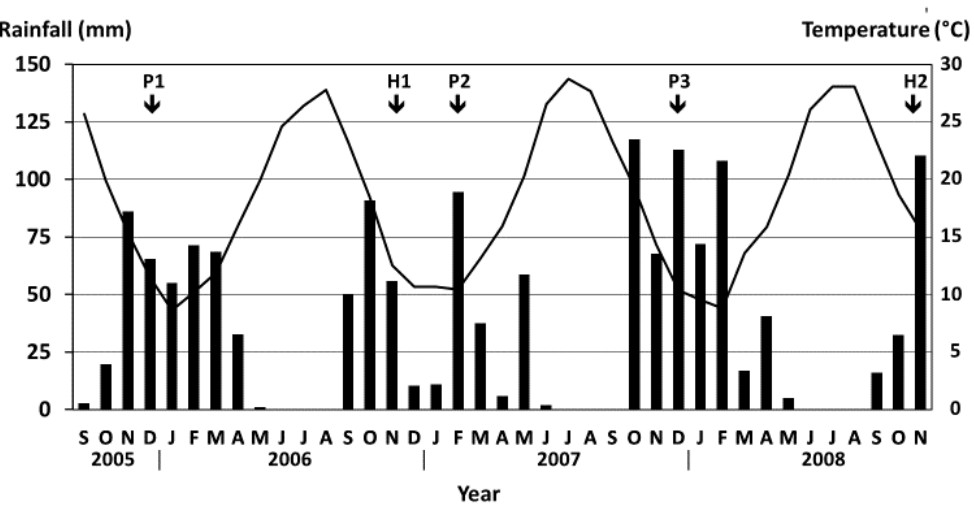

**Figure 3.** Average mean daily temperature and total rainfall per months in the three cover-crop growing seasons monitored. P1, P2 and P3 indicate planting dates for cover crops; H1 and H2 indicate the two olive harvests/olive fruit yield assessment dates in 2006 and 2008.

Before the start of the experiment the orchard was fertilised every second autumn by applying sheep manure at the rate of 20 m³/ha, and the establishment of vetch cover crops from untreated seed sown immediately after sheep manure was incorporated using the same chisel plough-type cultivator as during the experimental period. Vetch cover crops were sown by hand and then incorporated as a green manure in spring, using the same rotavator as during the experimental period. Olive tree canopy management was carried out annually after harvest, between January and March using standard pruning protocols, which involved a 'heavier' pruning protocol after harvesting a table olive crop, and a

'light' pruning in the year between the main fruiting seasons. The orchard was irrigated with drip irrigation during the dry season (May till October) and 150 m³ ha⁻¹ of water were applied every 15 days.

Cover crop, hand-sowing, irrigation and pruning methods remained the same throughout the 3-year experimental period. In the cropping seasons 2005/2006 and 2007/2008, fertilisation and sowing were performed at the beginning of December while in the non-cropping season (2006/2007) they were performed at the end of January due to dry conditions in December 2006 and January 2007 (Figure 3). Seed rates and fertility inputs are shown in Table 1.

*2.4. Leaf Analysis*

Leaf samples from olive trees were taken two times every year, in June and October. Each leaf sample comprised 400 healthy, mature leaves collected from the middle portion of bearing and non-bearing shoots from last season's growth, approximately 2 m above the soil surface, at the four cardinal points from the 2 middle trees of each plot. Entire, healthy, and mature leaves were collected, and immediately (within 1 h) transported to the laboratory, washed with deionised water and dried in a Memmert U15 forced-air oven (Memmert GmbH & Co. KG, Schwabach, Germany) at 60 °C for 48 h. Dry leaf samples were ground to a state of powder, with a sample mill "Cemotec 1090" (Foss A/S, Hilleroed Denmark). After leaf tissue samples used for mineral composition analyses were taken, the remaining tissue was placed into plastic jars and stored at room temperature until required.

**Table 1.** Seed rates and fertility inputs in different cover crops.

| Cover Crop | Seed Rates (kg/ha) | Fertilisation | | | |
| | | 2005/2006 | 2006/2007 | | 2007/2008 |
| | | Sheep Manure (m³/ha) | Agrobiosol (kg/plot) | Patentkali (kg/plot) | Sheep Manure (m³/ha) |
|---|---|---|---|---|---|
| 1 Vetch (−R) | 150 | 10 | 24 | 12 | 10 |
| 2 Mixture | 200 | 10 | 24 | 12 | 10 |
| *Vetch* | *120* | | | | |
| *Peas* | *50* | | | | |
| *Barley* | *30* | | | | |
| 3 Vetch (+R) | 150 | 10 | 48 | 12 | 20 |
| 4 Medicago | 160 | 10 | 24 | 12 | 10 |

−R, without *Rhizobium* inoculant; +R, with *Rhizobium* inoculants.

A 200 mg milled sample was weighed in Teflon vessels of a microwave digestion unit (CEM-Mars EXPRESS, Matthews, USA) and then 2 mL of $H_2O_2$ and 5 mL of $HNO_3$ were added to each sample and digested for 25 min at 1200 W in a microwave closed digestion unit. Digested samples were filtered through blue ribbon filter paper and the filtrate was collected in plastic graduated tubes and mixed with 20 mL of MilliQ water. Extracts were analysed for nitrate and mineral macro- and micronutrients with a simultaneous ICAP-OES (inductively coupled argon plasma optical emission spectrometer) equipped with a CCD detector (Varian-VistaPRO, Varian Inc., Palo Alto, USA). The instrument was calibrated with a mixed standard prepared in the same matrix as used for the plant samples (i.e., 2:5:13 ratio of $H_2O_2$: $HNO_3$: $H_2O$). NIST-1567a (Wheat Flour) and NIST-1547 (Peach Leaves) was used as a quality control sample with every 40 samples.

Milled leaf samples were analyzed for total nitrogen (N) using a LECO-N analyser using a standard protocol (Form No. 203-821-273) provided by the manufacturer (LECO corporation, St Joseph, USA). A total of 0.2 g of milled leaf sample was weighed into tin foil cups. The tin foils were wrapped and placed into the sample carousel of the

instrument. Instrument calibration was performed as outlined in the operator's instruction manual. Drift correction was performed daily, and conditions of combustion and afterburner temperatures were set to 950 °C and 850 °C, respectively, during each analysis run. In each batch of 60 samples a standard reference material from NIST (National Institute of Standards and Technology, Gaithersburg, USA) was used as a quality control sample along with a blank sample.

### 2.5. Assessment of Cover-Crop and Oxalis Establishment/Density

Assessments were carried out to estimate the number of plants per m² in each plot according to a year, in January, February/March and in April, just before incorporation of cover crops by ploughing (to 20 cm depth). Assessments were carried out using the methods described by Hodgson et al. [10] and Critchley and Poulton [11] which was based on counting plants within a 0.5 m × 0.5 m fixed quadrat.

### 2.6. Assessment of Nodulation in Legume Plants

Legume root nodulation was assessed each year in April, just before incorporation of cover crops. Twelve legume plants per plot were sampled at random after irrigation was applied for 30 min via the drip irrigation systems to allow easy removal of soil and roots. Roots were washed gently in water by hand and then placed onto a white paper, and all nodules on the recovered root system were counted. Five nodules from each plant were selected at random and used for size determination, which was performed using a digital calliper (Electronic Digital Callipers) and measuring nodule width and length. The same 5 nodules were then cut in half and all nodules showing pink color were assessed as being active, while nodules without colour were recorded as inactive.

### 2.7. Olive Yield Assessment

Olive yield assessments were carried out in cropping seasons 2006/2007 and 2008/2009. Due to the biennial fruiting no harvest assessment was possible in the cropping season 2007/2008. The two middle trees in each plot were harvested separately by hand, leaves were removed, and olive fruits were weighted. A sample of 400 olive fruits from each tree was taken (two samples per plot) and size (length and width) was measured using a digital calliper (Electronic Digital Callipers). The weight of 100 fruits, the weight of 100 stones (after manual removal of the pulp) and the maturity index were then determined on a subsample of 100 fruits. Maturity indexes were determined based on the method described by Uceda and Frias [12].

### 2.8. Olive Fruit Fly Sampling and Fruit Infestation Estimation

Estimations of fruit infestation by olive flies were carried out on the two trees (6 m apart) in the centre of the cover crop treatment plots. A total of 120 fruits from the two trees were examined for active and non-active infestation involving egg punctures, alive and dead eggs, and larvae. Estimations were carried out every two weeks from the 1st of July until the 15th of November in 2006 and 2008.

### 2.9. Statistical Analyses

The effects and interactions between factors on measured parameters were assessed by analysis of variance (ANOVA) derived from linear mixed-effects (LME) models [13] by using the 'nlme' package in R [14]. The hierarchical nature of the design was reflected in the random error structures that were specified as farm/year. The normality of the residuals of all models was tested using quantile–quantile (QQ) plots. Real means and standard errors of means were generated by using the 'tapply' function in R.

## 3. Results

### 3.1. Effect of Different Cover Crops on Vetch and Oxalis Establishment and Population Density

Cover crops established from *Rhizobium*-inoculated and untreated vetch seed and vetch sown as a mixture with barley and peas resulted in the satisfactory establishment of vetch (> 130 plants/m²) (Table 2). However, establishment of the native *Medicago* species resulted in very poor establishment (9 plants/m²) when compared to vetch cover crops established with untreated seed (156 plants/m²) (Supplementary Table S1).

The establishment of *Oxalis pes-caprae* was significantly higher in *Medicago* plots compared with vetch plots established with untreated seed (Supplementary Table S1). Additionally, *Oxalis* establishment was significantly lower in plots with cover crops established with (i) *Rhizobium*-inoculated vetch seed and (ii) a vetch/barley/pea seed mixture when compared to plots in which untreated vetch seeds were sown (Table 2).

ANOVA also detected significant main effects of growing season/year and assessment months on the density of both vetch and *Oxalis pes-caprae* and significant interactions between cover crop, year, and the months in which assessments were carried out (Table 2; Supplementary Table S2). The three-way interactions for both *Vicia sativa* and *Oxalis* density were therefore analysed in more detail (Figure 3; Supplementary Figure S1).

When interpreting the results of the interactions it is important to consider that cover crops were sown in October in the two olive harvest seasons (2005/2006 and 2007/2008) while in the non-harvest growing season (2006/2007) the planting of cover crops was delayed until the end of January 2007, due to a lack of sufficient rainfall in December 2006 and the first 3 weeks of January 2007 (Figure 3). Most importantly, this may explain why (i) vetch plant establishment in January was only detected in 2006 and 2008, but not 2007 and (ii) the overall higher Oxalis density in January 2007 compared to 2006 and 2007 (Figure 3; Supplementary Figure S1).

When the three vetch-based cover crop treatments were compared, vetch density in cover crops established from untreated *V. sativa* seeds was significantly higher when (i) compared with both other vetch-based cover crops in January 2006 and 2008, and February 2008 and (ii) compared with the cover crop establish from the seed mixture in February/March 2007 and April 2008 (Figure 3; Supplementary Figure S1). In contrast, vetch density in cover crops established from untreated *V. sativa* seeds was significantly lower compared with the two other vetch-based cover crop treatments in February/March 2006, April 2007 (Figure 3; Supplementary Figure S1).

When the effect of the three vetch-based cover crop treatments on *Oxalis* plant density was compared, significantly lower *Oxalis* density in plots with cover crops established from *Rhizobium*-inoculated compared with untreated seed were only detected in January 2006 and 2007 (Figure 3; Supplementary Figure S1). In contrast, compared to cover crops established with untreated vetch seed, the use of a vetch/barley/pea mixture resulted in a lower *Oxalis* density only in January 2007, and in both January and February 2008. It is important to note that *Oxalis* density in all plots with was significantly lower in 2008 than 2006 (Figure 3; Supplementary Figure S1).

**Table 2.** Effect of planting vetch (Vicia sativa) on its own (either inoculated and non-inoculated) or in a mixture with peas (Pisum sativum) and barley (Hordeum vulgare) on the relative establishment and development of vetch (Vicia sativa) and the weed Oxalis (Oxalis pes-caprae). Values shown are main effect means ±SE.

| Factor | | Vicia Density (Plants/m²) | Oxalis Density (Plants/m²) |
|---|---|---|---|
| Year | 2006 | 200 ± 6 **a** | 255 ± 8 **a** |
| | 2007 | 109 ± 6 **b** | 147 ± 12 **b** |
| | 2008 | 119 ± 4 **b** | 74 ± 2 **c** |
| Months | January | 146 ± 8 **b** | 258 ± 11 **a** |
| | February/March | 174 ± 5 **a** | 144 ± 7 **b** |
| | April | 108 ± 5 **c** | 74 ± 5 **c** |

| | | | | |
|---|---|---|---|---|
| Cover crop | *V. sativa* (−R) | 156 ± 6 **a** | 186 ± 11 **a** |
| | Mixture | 134 ± 6 **b** | 134 ± 8 **b** |
| | *V. sativa* (+R) | 138 ± 6 **b** | 156 ± 8 **b** |
| **ANOVA results** (*p*-values) | | | |
| **Main effects** | | | |
| Year (Y) | | <0.0001 | <0.0001 |
| Month (M) | | <0.0001 | <0.0001 |
| Cover crop treatment (T) | | 0.001 | <0.0001 |
| **Interactions** | | | |
| Y × M | | <0.0001 | <0.0001 |
| Y × T | | <0.0001 | <0.0001 |
| M × T | | <0.0001 | <0.0001 |
| Y × M × T | | <0.0001 [1] | <0.0001 [1] |

Values shown are main effect means (±SE); Means within the same column and for the same factor with the same letter are not significantly different (*p* < 0.05) according to THSD test; −R, without Rhizobium inoculant; +R, with Rhizobium inoculants; [1], see Figure 4 for interaction means.

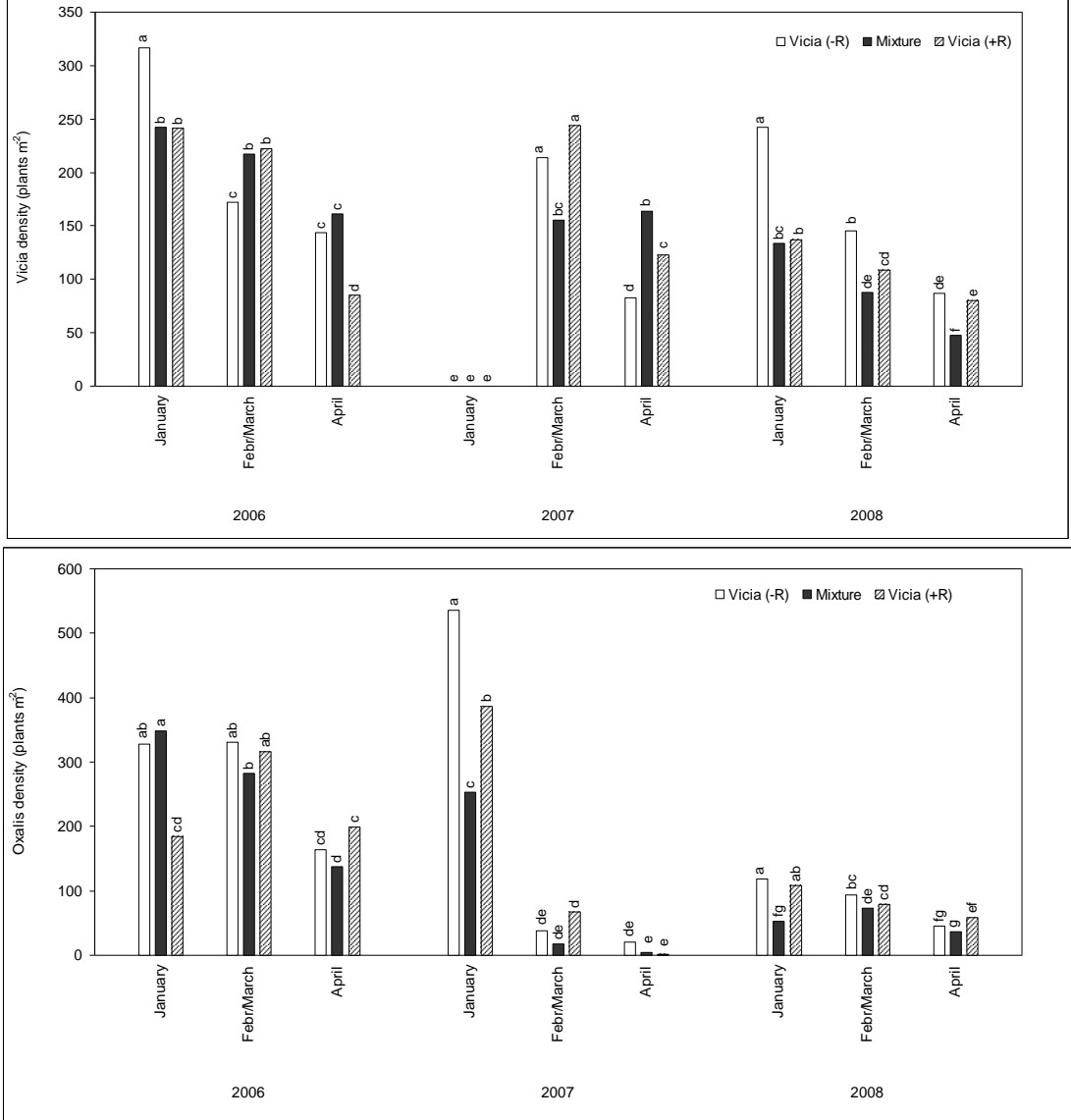

**Figure 4.** Interaction means for the effect of contrasting cover crops on vetch and Oxalis density in different months in the three experimental years. In each graph bars with different letters for the same year are significantly different according to THSD tests (*p* < 0.05).

### 3.2. Effect of Using Rhizobium Seed Inoculum on Vetch Establishment and Nodulation

Significant main effects of growing season/year were detected for the number of nodules per vetch plant and the mean size of nodules, and there was a trend ($0.1 > p > 0.05$) towards a significant effect of growing season for the number of active nodules per m² (Table 3). However, there was no significant main effect, or interactions with, *Rhizobium* inoculation on any of the nodulation parameters assessed (Table 3).

**Table 3.** Effect of year and *Rhizobium* seed treatment on vetch root nodulation/nitrogen fixation capacity related parameters. Values shown are main effect means ± SE.

| Factor | Total Number of Plants/m² | Total Number of Nodules/Plant | Mean Size of Nodules (mm) | Proportion of Active Nodules (%) | Total Number of Active Nodules/m² |
|---|---|---|---|---|---|
| **Year** | | | | | |
| 2006 | 101 ± 18 | 37 ± 2 **b** | 1.7 ± 0.2 **b** | 69 ± 12 | 2379 ± 341 **ab** |
| 2007 | 103 ± 16 | 50 ± 3 **a** | 1.1 ± 0.1 **c** | 66 ± 8 | 3019 ± 294 **a** |
| 2008 | 84 ± 6 | 43 ± 4 **ab** | 2.0 ± 0.1 **a** | 54 ± 6 | 1944 ± 269 **b** |
| *Rhizobium* **Seed Treatment** | | | | | |
| With | 92 ± 12 | 44 ± 4 | 1.6 ± 0.1 | 63 ± 8 | 2397 ± 290 |
| Without | 98 ± 12 | 43 ± 2 | 1.6 ± 0.2 | 62 ± 6 | 2510 ± 265 |
| **ANOVA Results** (*p*-values) | | | | | |
| **Main Effects** | | | | | |
| Year (Y) | NS | 0.0488 | < 0.0001 | NS | **T** |
| *Rhizobium* seed treatment (R) | NS | NS | NS | NS | NS |
| **2-Way Interactions** | | | | | |
| Y × R | NS | NS | NS | NS | NS |

Values shown are main effect means ± SE; means within the same column and for the same factor with the same letter are not significantly different (*p < 0.05*) according to THSD test. NS, not significant (*p < 0.1*); T, trend (*0.1 > p < 0.05*).

### 3.3. Effect of Different Cover Crops on Leaf Mineral Concentrations in Olive Leaves

Leaf mineral nutrient concentrations were compared to estimate the relative supply of mineral nutrients to olive trees in (i) plots sown with different cover crops and (ii) different years (2006, 2007 and 2008) and on (iii) two different sampling dates in each growing season (in June after incorporation of cover crops and 4 months later in October at the end of olive fruit development) (Tables 4 and 5).

**Table 4.** Effect of, and interaction between, (**a**) year (2006, 2007 or 2008), (**b**) cover crop treatment (vetch without *Rhizobium* inoculant, mixture, and vetch with *Rhizobium* inoculant or *Medicago*) and (**c**) sampling date (June or October) on the % concentration of macronutrients of olive leaves.

| Factor | | N (%) | NO₃ (%) | K (%) | P (%) | Mg (%) | Ca (%) | S (%) | Na (%) |
|---|---|---|---|---|---|---|---|---|---|
| **Year** | 2006 | 1.33 **b** (±0.02) | 0.096 **b** (±0.004) | 0.66 **c** (±0.01) | 0.073 **b** (±0.002) | 0.234 **a** (±0.007) | 1.92 **a** (±0.04) | 0.146 **a** (±0.003) | 0.0088 **a** (±0.0005) |
| | 2007 | 1.72 **a** (±0.03) | 0.111 **a** (±0.003) | 0.99 **a** (±0.02) | 0.096 **a** (±0.002) | 0.199 **b** (±0.003) | 1.20 **b** (±0.03) | 0.143 **ab** (±0.002) | 0.0076 **b** (±0.0003) |
| | 2008 | 1.37 **b** (±0.01) | 0.106 **a** (±0.003) | 0.88 **b** (±0.01) | 0.078 **b** (±0.002) | 0.234 **a** (±0.004) | 1.91 **a** (±0.03) | 0.140 **b** (±0.002) | 0.0070 **b** (±0.0002) |
| **Cover crop treatment** | *V. sativa* (−R) | 1.46 (±0.04) | 0.103 (±0.003) | 0.81 (±0.03) | 0.081 (±0.003) | 0.222 (±0.007) | 1.70 (±0.09) | 0.144 (±0.002) | 0.0080 (±0.0003) |
| | Mixture | 1.47 (±0.04) | 0.102 (±0.004) | 0.85 (±0.03) | 0.081 (±0.003) | 0.223 (±0.005) | 1.66 (±0.08) | 0.141 (±0.003) | 0.0082 (±0.0006) |
| | *V. sativa* (+R) | 1.49 (±0.05) | 0.104 (±0.004) | 0.84 (±0.04) | 0.083 (±0.003) | 0.223 (±0.008) | 1.71 (±0.09) | 0.145 (±0.002) | 0.0076 (±0.0004) |
| | Medicago | 1.47 (±0.04) | 0.108 (±0.004) | 0.86 (±0.03) | 0.085 (±0.003) | 0.222 (±0.007) | 1.64 (±0.07) | 0.142 (±0.002) | 0.0074 (±0.0005) |
| **Sampling date** | June | 1.51 (±0.04) | 0.112 (±0.003) | 0.91 (±0.03) | 0.086 (±0.002) | 0.205 (±0.003) | 1.55 (±0.06) | 0.141 (±0.002) | 0.0078 (±0.0003) |
| | October | 1.44 (±0.02) | 0.097 (±0.002) | 0.78 (±0.02) | 0.079 (±0.002) | 0.239 (±0.005) | 1.80 (±0.05) | 0.145 (±0.002) | 0.0079 (±0.0003) |
| **ANOVA results (p-values)** | | | | | | | | | |
| **Main effects** | | | | | | | | | |
| Year (Y) | | **< 0.0001** | **0.0015** | **< 0.0001** | **< 0.0001** | **< 0.0001** | **< 0.0001** | **0.0015** | **0.0058** |
| Cover crop treatment (C) | | NS | NS | NS | *T* | NS | NS | NS | NS |
| Sampling date (D) | | **< 0.0001** | **< 0.0001** | **< 0.0001** | **< 0.0001** | **< 0.0001** | **< 0.0001** | **0.0015** | NS |
| **Interactions** | | | | | | | | | NS |
| Y × C | | NS | NS | NS | NS | NS | NS | **T** | NS |
| Y × D | | **< 0.0001** [1] | NS | **< 0.0001** [1] | **< 0.0001** [1] | **0.0318** [1] | **0.0015** [1] | **< 0.0001** [1] | NS |
| C × D | | NS | NS | NS | NS | NS | NS | NS | NS |
| Y × C × D | | NS | NS | NS | NS | NS | NS | NS | NS |

Values shown are main effect means (±SE); Means within the same column and for the same factor with different letters are significantly different (*p* < 0.05) according to THSD test; −R, without *Rhizobium* inoculant; +R, with *Rhizobium* inoculants. See Figure 5 for interaction means. NS, not significant (*p* < 0.1); T, trend (*0.1 > p < 0.05*) ; [1], see Figure 5 for interaction means.

Significant main effects of cover crop treatment were only detected for B and Mo levels in leaves, with the *Medicago* treatment (which due to the poor establishment of *Medicago* can also be considered a no cover crop control treatment) resulting in the highest B and Mo levels in olive tree leaves (Tables 4 and 5).

Significant main effects of year and sampling date were detected for all macro and micronutrients and a significant main effect of year was also detected for Na (Tables 4 and

5). Concentrations of N, NO₃ K, P, Fe, Zn, Cu, B and Mo were higher in June after incorporation of cover crops in April/May, while Mg, Ca, S and Mn concentrations were higher in October (Tables 4 and 5).

**Table 5.** Effect of, and interaction between (**a**) year (2006, 2007 or 2008), (**b**) cover crop treatment (vetch without *Rhizobium* inoculant, mixture, vetch with *Rhizobium* inoculant or *Medicago*) and (**c**) sampling date (June or October) on the concentration of micronutrients of olive leaves.

| Factors | | Fe (mg kg⁻¹) | Mn (mg kg⁻¹) | Zn (mg kg⁻¹) | Cu (mg kg⁻¹) | B (mg kg⁻¹) | Mo (mg kg⁻¹) |
|---|---|---|---|---|---|---|---|
| **Year** | 2006 | 131 ± 4 **b** | 49.6 ± 3.0 **b** | 15.4 ± 0.4 **a** | 3.6 ± 0.1 **b** | 17.6 ± 0.3 **b** | 0.285 ± 0.009 **a** |
| | 2007 | 81 ± 3 **c** | 35.3 ± 1.8 **c** | 15.6 ± 0.5 **a** | 4.6 ± 0.2 **b** | 19.9 ± 0.5 **a** | 0.186 ± 0.007 **b** |
| | 2008 | 158 ± 4 **a** | 59.6 ± 2.6 **a** | 13.8 ± 0.3 **b** | 3.1 ± 2.5 **a** | 17.3 ± 0.3 **b** | 0.160 ± 0.007 **c** |
| Cover crop | *V. sativa* (-R) | 127 ± 7 | 48.3 ± 3.7 | 15.0 ± 0.6 | 13.2 ± 3.2 | 17.1 ± 0.4 b | 0.209 ± 0.015 ab |
| | Mixture | 123 ± 8 | 48.9 ± 3.4 | 15.3 ± 0.6 | 11.9 ± 2. | 18.7 ± 0.5 a | 0.185 ± 0.012 b |
| | *V. sativa* (+R) | 122 ± 7 | 47.5 ± 3.6 | 14.2 ± 0.4 | 13.6 ± 3.3 | 18.1 ± 0.4 ab | 0.221 ± 0.016 ab |
| | Medicago | 120 ± 8 | 48.0 ± 3.5 | 15.3 ± 0.5 | 13.5 ± 3.3 | 19.2 ± 0.5 a | 0.226 ± 0.013 a |
| Sampling date | June | 131 ± 5 | 46.0 ± 2.5 | 15.6 ± 0.4 | 17.5 ± 2.8 | 19.0 ± 0.4 | 0.218 ± 0.010 |
| | October | 115 ± 5 | 50.4 ± 2.4 | 14.3 ± 0..3 | 8.6 ± 1.1 | 17.6 ± 0.3 | 0.203 ± 0.010 |
| **ANOVA results** (*p*-values) | | | | | | | |
| **Main effects** | | | | | | | |
| Year (Y) | | < 0.0001 | < 0.0001 | < 0.0001 | < 0.0001 | < 0.0001 | < 0.0001 |
| Cover crop treat (C) | | NS | NS | T | NS | 0.0002 | 0.0011 |
| Sampling date (D) | | < 0.0001 | 0.0024 | 0.0004 | < 0.0001 | 0.0001 | 0.0490 |
| **Interactions** | | | | | | | |
| Y × C | | NS | NS | T | NS | NS | NS |
| Y × D | | NS | NS | < 0.0001 | < 0.0001 | < 0.0001 | NS |
| C × D | | NS | NS | NS | NS | NS | NS |
| Y × C × D | | NS | NS | 0.0178 | NS | NS | NS |

Values shown are main effect means ±SE; means within the same column and for the same factor with different letters are significantly different (*p < 0.05*) according to THSD test; -R, without *Rhizobium* inoculant; +R, with *Rhizobium* inoculants. NS, not significant (*p < 0.1*); T, trend (*0.1 > p < 0.05*).

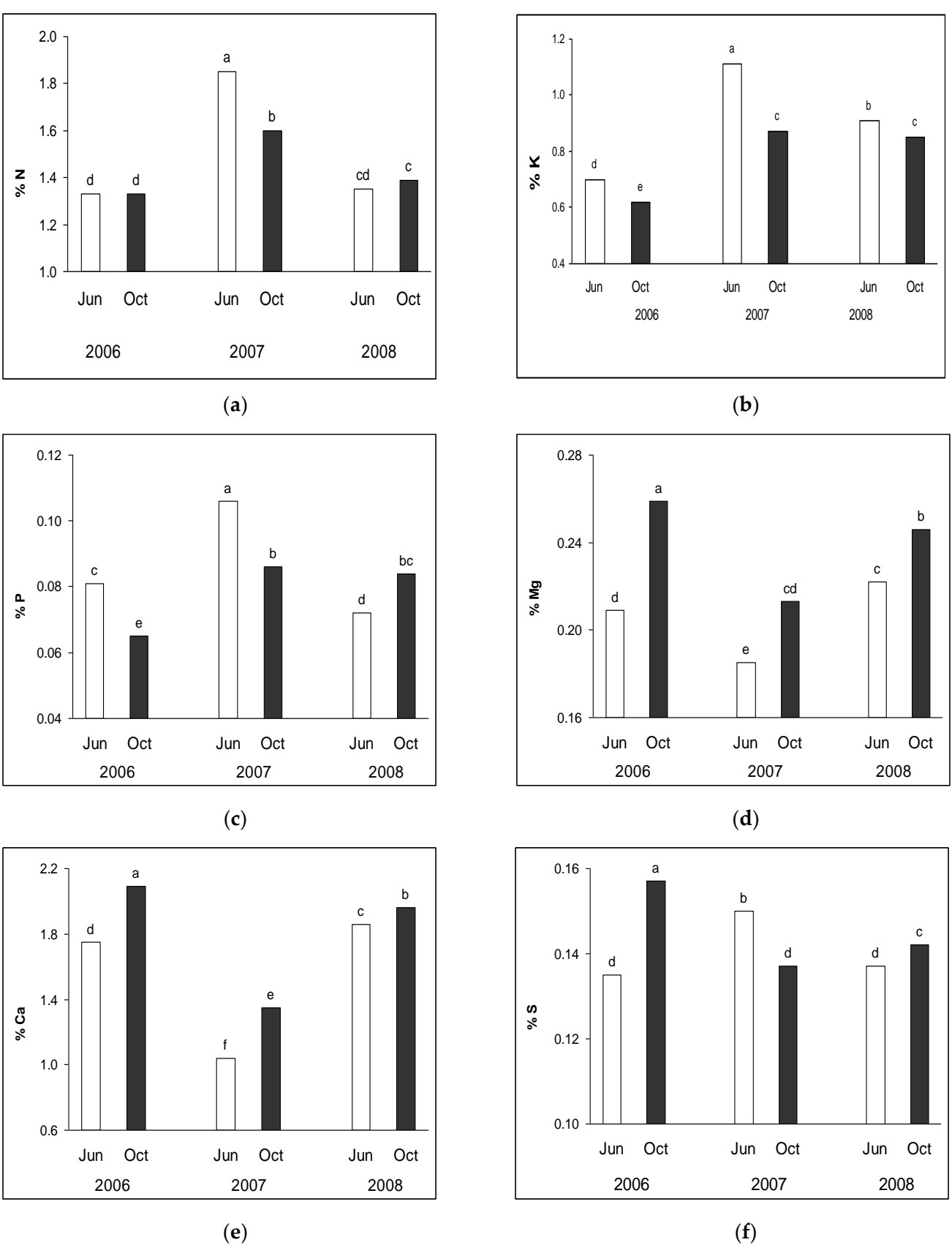

**Figure 5.** Effect of year and sampling date on (**a**) N, (**b**) K, (**c**) P, (**d**) Mg, (**e**) Ca and (**f**) S concentrations of olive leaves. Bars with the same letter within each graph are not significantly different according to THSD tests (*p < 0.05*); Jun, June; Oct, October. Bars are interaction means.

Concentrations of N, nitrate, K, P, and B were significantly higher while concentrations of Ca, Mg, Fe, and Mn were lower in the non-harvest year (2007) when compared to the two harvest years (2006 and 2008) (Tables 4 and 5).

For a wide range of macro- and micronutrients ANOVA also detected significant interaction between year and date (Tables 4 and 5; Figure 5), and there was a significant interaction between year, cover crop and sampling date for Zn (Table 5).

### 3.4. Effect of Cover Crops on Table Olive Yield and Quality Parameters

There was no significant main effect of cover crop treatment, although it should be noted that numerically fruit yields were higher (between 12 and 21%) with the *Medicago* treatment (which resulted in very poor cover crop establishment; Supplementary Table S1) when compared to the three vetch-based cover crop treatments (Table 6).

**Table 6.** Effect of, and interaction between (**a**) cover crop treatments and (**b**) growing season on olive yield parameters. Values shown are main effect means ±SE.

| Factor | | Yields (kg tree$^{-1}$) | Weight of 100 Fruits (g) | Weight of 100 Stones (g) | Pulp/Stone Ratio | Maturity Index |
|---|---|---|---|---|---|---|
| | *V. sativa* (−R) | 38.2 ± 3.7 | 422 ± 24 [1] | 51.2 ± 1.9 | 7.4 ± 0.6 | 4.7 ± 0.2 [1] |
| | Mixture | 45.2 ± 6.4 | 422 ± 26 [1] | 51.4 ± 1.1 | 7.2 ± 0.5 | 4.8 ± 0.3 [1] |
| **Cover Crop** | *V. sativa* (+R) | 44.3 ± 4.1 | 385 ± 21 [1] | 48.8 ± 1.4 | 7.1 ± 0.6 | 4.4 ± 0.2 [1] |
| | Medicago | 50.2 ± 4.4 | 397 ± 27 [1] | 51.1 ± 0.9 | 6.8 ± 0.6 | 4.4 ± 0.2 [1] |
| Growing | 2005/2006 | 55.1 ± 2.9 | 327 ± 7 | 53.1 ± 0.7 | 5.2 ± 0.1 | 3.9 ± 0.1 |
| Season | 2007/2008 | 34.3 ± 2.9 | 483 ± 21 | 48.2 ± 1.0 | 9.1 ± 0.2 | 5.2 ± 0.1 |
| **ANOVA results** (*p*-values) | | | | | | |
| **Main effects** | | | | | | |
| Cover crop treatment (C) | | NS | 0.0304 [1] | NS | NS | 0.0443 [1] |
| Growing season (S) | | < 0.0001 | < 0.0001 | < 0.0001 | < 0.0001 | < 0.0001 |
| **Interaction** | | | | | | |
| C × S | | NS | 0.0118 [1] | 0.0042 [2] | NS | NS |

−R, without *Rhizobium* inoculant; +R, with *Rhizobium* inoculants; NS, not significant (*p* < 0.1); [1], no significant differences between main effect means for different cover crop treatments could be identified by THSD tests; [2], see Table 7 for interaction means ±SE.

However, ANOVA detected small but significant main effects of cover crop for the 100-fruit weight and the maturity index of the olives, which were slightly higher in plots where untreated vetch seed or a vetch/barley/pea seed mixture was established (Table 6).

Significant main effects of the growing season were detected for fruit yield (higher in the 2005/2006 harvest season), the 100-fruit weight (higher in 2007/2008 harvest season), the 100-stone weight (higher in the 2005/2006), the pulp/stone ratio (higher in 2007/2008 harvest season) and the maturity index (higher in 2007/2008 harvest season) (Table 6).

Significant interactions between the cover crop and growing season were detected for the 100-fruit and the 100-stone weights (Table 6). When the two-way interactions were further investigated, untreated vetch plots produced the highest and *Medicago* plots the lowest 100-fruit and 100-stone weights in the 2005/2006 season, while the cover crops mixture plots produced the highest and *Rhizobium*-inoculated vetch plots the lowest 100-fruit and 100-stone weights in the 2007/2008 season (Table 7).

**Table 7.** Effect of different cover crop treatment on the 100-fruit and 100-stone weights of olives in the two growing seasons.

| Growing Season | Cover Crop Treatment | Weight of 100 Fruits (g) | Weight of 100 Stones (g) |
|---|---|---|---|
| | *V. sativa* (−R) | 350 ± 12 **a** | 56.2 ± 1.9 **a** |
| **2005/2006** | Mixture | 337 ± 17 **a** | 51.8 ± 1.0 **b** |
| | *V. sativa* (+R) | 327 ± 11 **ab** | 53.3 ± 0.7 **ab** |
| | Medicago | 295 ± 6 **b** | 51.5 ± 1.1 **b** |
| | *V. sativa* (−R) | 484 ± 28 **a** | 46.8 ± 2.3 **ab** |
| 2007/2008 | Mixture | 507 ± 24 **a** | 51.1 ± 2.1 **a** |
| | *V. sativa* (+R) | 443 ± 27 **b** | 44.2 ± 1.5 **b** |
| | Medicago | 499 ± 8 **a** | 50.8 ± 1.5 **a** |

Values shown are interaction means ±SE; means for the same growing season within the same column with different letters are significantly different ($p < 0.05$) according to THSD test; −R, without *Rhizobium* inoculant; +R, with *Rhizobium* inoculants.

Fruit size, which is an important quality parameter that affects marketable yield and profitability of production, was therefore also recorded in both growing seasons (Table 8).

Very highly significant main effects of cover crop treatment and growing season were detected (Table 8). Overall, the highest and lowest mean olive fruit length, width and size were recorded in plots in which cover crops were established from a vetch/barley/pea seed mixture and *Rhizobium*-inoculated vetch seed, respectively (Table 8). Additionally, overall, fruits were ~15% larger in the 2007/2008 growing season compared with the 2005/2006 growing season (Table 8).

ANOVA also detected very highly significant interactions between cover crop and growing season (Table 8). When interactions were further investigated, in the 2005/2006 season cover crops established from untreated vetch and the vetch/barley seed mixture resulted in the largest fruit, while the smallest fruit were found in *Medicago* plots. In contrast, in the 2007/2008 season, the largest mean fruit size was recorded in *Medicago* plots, while the lowest mean fruit size was found in plots established from *Rhizobium*-inoculated vetch seed (Table 9).

Olive fruit fly infestation also affects the quality of table olives, and there is a very low tolerance for fruit fly lesions by table olive processors. Olive fly fruit infestation was therefore also assessed in growing season 2007/2008 but remained very low (< 1%) and no effect of cover crop treatment could be detected (individual data not shown).

**Table 8.** Effect of, and interaction between (**a**) different cover crop treatment (vetch without *Rhizobium* inoculum, mixture, vetch with *Rhizobium* inoculum, *Medicago*) and (**b**) growing season (2005/2006 or 2007/2008) on olive fruit dimensions.

| Factor | | Olive Fruit Length (mm) | Olive Fruit Width (mm) | Olive Fruit Size (mm²) |
|---|---|---|---|---|
| | *V. sativa* (−R) | 24.41 ± 0.05 **b** | 16.67 ± 0.02 **a** | 410 ± 1 **b** |
| Cover crop treatment | Mixture | 24.77 ± 0.03 **a** | 16.65 ± 0.03 **a** | 415 ± 1 **a** |
| | *V. sativa* (+R) | 24.20 ± 0.03 **c** | 16.17 ± 0.07 **b** | 393 ± 2 **d** |
| | Medicago | 24.47 ± 0.03 **b** | 16.25 ± 0.03 **b** | 401 ± 1 **c** |
| Growing season | 2005/2006 | 24.37 ± 0.02 | 15.26 ± 0.02 | 375 ± 1 |
| | 2007/2008 | 24.55 ± 0.03 | 17.56 ± 0.03 | 434 ± 1 |
| **ANOVA results** (*p*-values) | | | | |
| **Main effects** | | | | |
| Cover crop treatment (C) | | **< 0.0001** | **< 0.0001** | **< 0.0001** |
| Growing season (S) | | **< 0.0001** | **< 0.0001** | **< 0.0001** |
| **Interactions** | | | | |
| C × S | | **< 0.0001** | **< 0.0001** | **< 0.0001** |

Values shown are main effect means ±SE; cover crop main effect means within the same column with different letters are significantly different ($p < 0.05$) according to THSD test; −R, without *Rhizobium* inoculant; +R, with *Rhizobium* inoculants; size, length × width.

**Table 9.** Effect of cover crops on olive fruit dimension in the two different growing seasons.

| Growing Season | Cover Crop Treatment | Olive Fruit Length (mm) | Olive Fruit Width (mm) | Olive Fruit Size (mm) |
|---|---|---|---|---|
| | *V. sativa* (−R) | 24.66 ± 0.04 **a** | 15.55 ± 0.03 **a** | 386 ± 1 **a** |
| 2005/06 | Mixture | 24.73 ± 0.04 **a** | 15.58 ± 0.04 **a** | 388 ± 1 **a** |
| | *V. sativa* (+R) | 24.34 ± 0.04 **b** | 15.14 ± 0.03 **b** | 371 ± 1 **b** |
| | Medicago | 23.79 ± 0.04 **c** | 14.79 ± 0.03 **c** | 354 ± 1 **c** |
| 2007/08 | *V. sativa* (−R) | 24.19 ± 0.08 **c** | 17.65 ± 0.03 **a** | 430 ± 2 **c** |
| | Mixture | 24.81 ± 0.04 **b** | 17.72 ± 0.03 **a** | 441 ± 1 **b** |
| | *V. sativa* (+R) | 24.07 ± 0.04 **c** | 17.19 ± 0.13 **b** | 415 ± 3 **d** |
| | Medicago | 25.14 ± 0.04 **a** | 17.71 ± 0.03 **a** | 447 ± 1 **a** |

Values shown are interaction means ±SE; means for the same growing season within the same column with different letters are significantly different ($p < 0.05$) according to THSD test; −R, without *Rhizobium* inoculant; +R, with *Rhizobium* inoculant; size, length × width.

## 4. Discussion

### 4.1. Effect of Cover Crops on Oxalis Establishment

*Oxalis pes-caprae* is a noxious invasive weed that propagates largely through its underground bulbs, and this is the main reason why it is so difficult to control or eradicate by mechanical weed control of hand-weeding as pulling up the plant, even with the roots, can leave some of the bulbs behind [15–17]. In intensive conventional olive groves Oxalis can be efficiently controlled by herbicides such as glyphosate, but all currently available herbicide products are prohibited in organic farming [15–17]. Oxalis establishment was shown to have a considerable impact on the diversity and ecosystem functions in olive groves [15–17].

Results from this study suggest that overall Oxalis establishment and plant density was reduced by establishing pure vetch or vetch/barley/pea cover crops, but also that both Oxalis and *V. sativa* plant density declined between January and April in each growing season. Oxalis is known be an anthropogenic, very fast-growing weed that emerges rapidly, especially when soils are ploughed after the first substantial rainfall in autumn [3,15,17,18].

Oxalis growth and competitiveness was also reported to be reduced by low temperatures and dry conditions [3,17] which may explain the high plant density in January and lower density detected in February/March and April.

Oxalis was shown to compete poorly with certain cover crops due to its shallow root system, especially with grasses such as barley, which was part of the species mixture used in this study [15,19–21]. This is likely to at least partially explain that the lowest Oxalis plant density were recorded when a vetch/barley/pea seed mixture was used as cover crop.

It is important to note that in January 2007 both plots sown with *Rhizobium*-inoculated vetch seed and the vetch/barley/pea seed mixture had a lower Oxalis density than plots in which untreated vetch seed were sown and that at the time of January assessment, vetch peas and barley had not emerged in 2008. The low rainfall between October and January in the 2006/2007 growing season is the most likely reason why Oxalis plant density on all three assessment dates was substantially lower in 2006/2007 compared with the 2005/2006 and the 2007/2008 season. However, competition for water, light or nutrients by the cover crop cannot explain the difference in Oxalis density in the 2006/2007 season,

but differences may have been due to residual effects of the cover crops grown in the previous (2005/2006) season and/or a direct effect the *Rhizobium* inoculum used.

It also remains unclear why Oxalis density was overall lower in the 2007/2008 when compared with the 2005/2006 season. However, it is more likely that this was due to a cumulative 'weed-suppressive' effect of establishing cover crops in three consecutive years, since the higher rainfall in October and November (= before planting of cover crops) in 2007 compared with 2005 would have provided more favourable climatic conditions for Oxalis establishment in the 2007/2008 cover crop growing season.

### 4.2. Effect of Rhizobium Inoculation on Soil Fertility and Crop Performance

The use of *Rhizobium* inoculation of legume seed was reported to increase symbiotic N-fixation by legume crops, soil fertility and yields of subsequent crops in arable production systems [4,7,22]. The finding that *Rhizobium* inoculation of vetch (*V. sativa*) seed resulted (a) in a reduction in vetch crop density, (b) no significant increase in N-fixation efficacy (based on parameters measured such as nodulation, size of nodules and proportion of active nodules) and (c) no significant increase in N-availability (assessed via leaf analysis), was therefore unexpected.

The absence of an effect on nodulation may be explained by the fact that vetch was used as a cover crop in previous growing seasons, resulting in the presence of sufficient natural inoculum in soils to facilitate optimum nodule development, as previously reported [4,7,23,24]. For example, Thies et al. reported as early as 1991 that the probability of enhancing yield with *Rhizobium* inoculum decreases dramatically in soils with a high indigenous *Rhizobium* population density [25]. Additionally, *Rhizobium* inoculum that have been developed for other agronomic and climatic conditions may not be as effective under dry Mediterranean conditions [26]. However, the lower vetch plant density in cover crops established from inoculated vetch seed, suggests that the *Rhizobium* inoculum had a negative effect on the germination and/or establishment of vetch plants. However, additional studies would be required to identify the mechanisms involved. Overall, the results suggest that there are no detectable agronomic benefits from using *Rhizobium* inoculum for vetch cover crop in organic olive orchards.

### 4.3. Effect of Different Cover Crop Treatments on Crop Performance

Since mineral nitrogen fertilizers are prohibited by organic farming standards the use of legume cover crops in organic is also recommended as a method to increase N-supply and balanced the ratio of plant available N:P:K in soils that receive regular inputs of animal manure ratio [1,2,4]. However, it is important to consider that the N-requirements of olive trees are relatively low (~50 kg N/ha) and that excessive N-fertilisation is known to result in reduced resistance against abiotic (e.g., frost) and biotic (e.g., olive leaf spot caused by *Spilocea oleagina*) stress [27].

The finding that olive yields of the table olive variety 'Kalamon' in plots with vetch cover crops were not significantly different to those recorded in *Medicago* plots (which had a very low *Medicago* plant density and can be considered a no-cover crop control treatment) was therefore not surprising. It is important to note that fruit yields of the variety Koroneki (which is the dominant variety used for olive oil production in Crete) were reported to be not significantly different, in fact numerically ~10% higher, in organic compared with conventionally managed fields in the Messara region [4,28].

In contrast, the finding of lower concentrations of B and Mo in olive leaves in plots with the three vetch-based cover crops compared with *Medicago* plots suggests that vetch-based cover crops reduced B and Mo availability to olive trees to a larger extent than the dominant weed species *Oxalis*. This view is supported by previous studies which showed that legumes have a relatively high B requirements and that adequate B availability is required for effective nitrogen fixation and nodulation in legume crops [29,30].

Boron deficiency is a common problem in some olive growing areas [2,5,6] and the lower B (and possibly Mo) availability may have contributed to the larger fruit size and

weight recorded in the second fruiting season for plots in which *Medicago* was used as cover crop.

However, in the first growing season non-inoculated *Vicia sativa* and the mixed cover crop resulted in the highest fruit yield and size, indicating that factors other than B and Mo supply did affect yield. It is interesting to note that *Rhizobium*-inoculated *Vicia sativa* cover crops not only resulted in lower *Vicia sativa* establishment, but also in the lowest fruit yield and size in both fruiting seasons. This indicates that *Vicia sativa* establishment is linked to performance, but the underlying mechanisms remain unclear, since no differences in mineral macronutrient supply could be detected between un-inoculated and inoculated *Vicia sativa* cover crops.

Changes in ground cover vegetation are also thought to affect natural enemy population and thereby pest infestations levels (in particular the olive fruit fly *Bactrocera oleae*), which in turn may affect olive fruit yield and quality parameters [2,31,32]. However, potential impacts of cover crops on olive fruit fly infestation could not be investigated in this study, since olive fruit fly infestations was efficiently controlled by mass-trapping in the commercial orchard used for the experiment [3]. However, it is important to note that a pilot study that investigated background invertebrate populations in cover crop plots in the 2006/2007 and 2007/2008 reported three-times higher *Hymenoptera* (an insect order which includes a range of natural enemy species) activity in plots with vetch/barley/pea cover crops compared with plots in which untreated vetch or *Medicago* seed were sown to establish cover crops [3].

### 4.4. Potential for Using a Native Legume Species as Cover Crop

The use of the native legume species *Medicago polymorpha* has been suggested as an alternative to the use of vetch, which is currently the main legume species used as cover crop in organic olive production in Crete [2,3].

Attempts to establish legume cover crops with *Medicago polymorpha* seed collected in olive groves in the Messara region failed in all three growing seasons. Previous studies suggest that this may have been caused by poor seed quality and/or dormancy [33–36] and this is supported by the finding that the used in experiments only had a germination rate of 55%).

### 4.5. Study Limitations

The main limitation of this study was that cover crop biomass assessments were not carried out, which would have been necessary to gain a more in-depth understanding of the effects of cover crops on weed suppression and nutrient supply to soils. Additionally, the low germination rate of the *Medicago* seed obtained by collection of seeds in olive orchards made it impossible to assess the potential of *Medicago* as a cover crop. These limitations should be considered in the design of future studies.

### 5. Conclusions

Results showed that, although *Medicago* establishment was very low (< 10 plants/m$^2$), fruit yields were numerically higher in the *Medicago* plots, where cover crop establishment was poor. This suggests that vetch-based cover crops and the use of *Rhizobium* seed inoculum had no positive effect on fruit yields. This conclusion is supported by the results of the olive leaf analyses which detected no significant differences in nitrogen and other mineral macro and micronutrient concentration between treatments, except for B and Mo.

The finding that B and Mo levels were lower in plots with vetch-based cover crops than the plots with poorly established *Medicago* cover crops, suggests that legumes may compete with olive trees, because both legumes and olive trees have a relatively high B-requirement [30,37,38].

Given the (a) additional cost of establishing cover crops and *Rhizobium* inoculum, (b) the lack of detectable benefits from the *Rhizobium* inoculum and (c) potential competition

for B from legume cover crops, it is important to advise organic farmers (a) against the use of *Rhizobium* inoculum and (b) to not establish legume-based cover crops every year. Based on the result reported here, the advice to farmers should be to establish cover crops only every 2–4 years, since this may significantly reduce the cost of production without affecting olive fruit yields and quality.

Future research should focus on the development of innovative ground cover management methods which encourage the establishment of native legume-rich plant communities and thereby reduce *Oxalis* density.

**Supplementary Materials:** The following are available online at https://www.mdpi.com/article/10.3390/agronomy12102523/s1, Supplementary Table S1. Effect of using seed of a commercial legume variety (*Vicia sativa*) and seed of a native legume species (*Medicago polymorpha*) collected in olive orchards on the relative establishment of the respective legume species and the weed *Oxalis pes-caprae*; Supplementary Table S2. Effect of *Rhizobium* inoculation on the establishment and development of vetch (*Vicia sativa*) and the weed Oxalis (*Oxalis pes-caprae*); Supplementary Figure S1. Effect of year, month, and cover crop treatment (with and without *Rhizobium* seed inoculation) on vetch and Oxalis density. Bars with the same letter for the same year are not significantly different according to THSD tests ($p < 0.05$); −R = without *Rhizobium* inoculant, +R = with *Rhizobium* inoculants. Values shown are interaction means ± SE.

**Author Contributions:** Conceptualisation, N.V., E.M.K. and C.L.; methodology, N.V., E.M.K. and C.L.; software, N.V. and L.R.; validation, E.M.K., and C.L.; formal analysis, N.V. and L.R.; investigation, N.V.; resources, N.V., E.M.K. and C.L.; data curation, N.V.; writing—original draft preparation, N.V. E.M.K. and C.L.; writing—review and editing, N.V., E.M.K., A.K., L.R. and C.L.; visualisation, N.V.; supervision, E.M.K. and C.L.; project administration, N.V., E.M.K. and C.L.; funding acquisition, N.V., E.M.K. and C.L. All authors have read and agreed to the published version of the manuscript.

**Funding:** This research was funded by a scholarship from the Greek State Scholarships Foundation (IKY) to Nikos Volakakis, the European Union Integrated project Quality Low Input Food (Grant number 506358) and by GSRT-Greece matching funds for European Projects 2019—IGIC—"Improvement of green infrastructure (GI) in agroecosystems: reconnecting natural areas by countering habitat fragmentation" (Grant number 80811).

**Data Availability Statement:** Data will be made available upon reasonable request by author Nikolaos Volakakis.

**Acknowledgments:** Authors would like to thank the olive grower that make available and allow the research work in the olive orchard and for his time and patience.

**Conflicts of Interest:** Authors N.V., E.K. and C.L. own and manage organic olive farms in Crete and author C.L. is a member of the UK soil association. The funders had no role in the design of the study; in the collection, analyses, or interpretation of data; in the writing of the manuscript; or in the decision to publish the results.

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
