# Peer review of "Effect of Different Cover Crops on Suppression of the Weed Oxalis pes-caprae L., Soil Nutrient Availability, and the Performance of Table Olive Trees ‘Kalamon’ cv. in Crete, Greece"

_agronomy, doi:10.3390/agronomy12102523_

Round 1

Reviewer 1 Report

In materials and methods, the Olive fruit fly sampling and fruit infestation was estimated, but the data was not showed in Results.

Author Response

We have described in the results section that olive fly infestation levels were very low (<1%) and that there were no significant differences between cover crop treatments with the following text:

“Olive fly fruit infestation also affects the quality of table olives, and there is a very low tolerance for fruit fly lesions by table olive processors. Olive fly fruit infestation was therefore also assessed in growing season 2007/2008 but remained very low (<1%) and no effect of cover crop treatment could be detected (individual data not shown). (lines 334-337)”

We feel it is unnecessary to present the detail of the results in a Table, because infestation levels were so low (<1%) and because there was no effect of cover crop treatment.

Reviewer 2 Report

The authors propose a manuscript titled “Effect of different cover crops on suppression of the weed Oxalis pes-caprae L., soil nutrient availability, and the performance of table olive trees Kalamon cv.; results from a 3-year field trial in Crete, Greece”.

I suggest the following changes:

Introduction: I suggest expanding this section, with the addition of more references

Results: Please add mean squares at the ANOVA table.

Conclusion: I suggest rewriting conclusion in order to present more clearly and in detail.

Author Response

Introduction: I suggest expanding this section, with the addition of more references

Authors’ response:

We have now replaced reference 7 (lines 518-520) with a more recent reference that describes/reviews the use of cover crops in arable systems.

We have also now revised and expanded the conclusion section and added additional references (references 37, 38 and 39) which describe the importance of B for both legume plants and olive trees (lines 461-464).

Results: Please add mean squares at the ANOVA table.

Authors’ response: this is not usually done/required in ANOVA tables (see for example the new reference 7 which was recently published in agronomy) and feel that describing the means, a measure of the variation (SE) in the Tables and experimental design/replication in the material and methods is sufficient.

Conclusion: I suggest rewriting conclusion in order to present more clearly and in detail.

Authors’ response:

We have now revised and expanded the conclusion section and provides three new references (37-39) in this section. We have also separated the text into several paragraphs. The conclusion section now reads:

“5. Conclusions

                Results showed that, although Medicago establishment was very low (<10plants/m2), fruit yields were numerically higher in the Medicago plots, where cover crop establishment was poor. This suggests that vetch-based cover crops and the use of Rhizobium seed inoculum had no positive effect on fruit yields. This conclusion is supported by the results of the olive leaf analyses which detected no significant differences in nitrogen and other mineral macro and micronutrient concentration between treatments, except for B and Mo.

                The finding that B and Mo levels were lower in plots with vetch-based cover crops than the plots with poorly established Medicago cover crops, suggests that legumes may compete with olive trees, because both legumes and olive trees have a relatively high B-requirement [37-39].

                Given the (a) additional cost of establishing cover crops and Rhizobium inocula, (b) the lack of detectable benefits from Rhizobium inoculum and (c) potential competition for B from legume cover crops, it is important to advise organic farmers (a) against the use of Rhizobium inoculum and (b) to not establish legume-based cover crops every year. Based on the result reported here, the advice to farmers should be to establish cover crops only every 2-4 years since this may significantly reduce cost of production without affecting olive fruit yields and quality.

                Future research should focus on the development of innovative ground cover management methods which encourage the establishment of native legume-rich plant communities and thereby reduce Oxalis density.” (lines 453-473)

Reviewer 3 Report

The manuscript ‘Effect of different cover crops on suppression of the weed Oxalis pes-caprae L., soil nutrient availability, and the performance of table olive trees Kalamon cv.; in Crete, Greece’ deals with the suppression of dominant weed species in vetch cultivation using various cover crops. The study is planned and presented in a good way and deserves publication. Minor work is required to bring the manuscript in publishable form.

The title must not contain results from a 3-year field trial

The abstract is fine. Just add the age of the orchard

The last 3 lines of introduction section must be added within the sentences

Please add the age of the orchard in MM section and write cultivar names in inverted commas.

The results of Medicago have been skipped in Tables and figures relating to Oxalis density? If it was for comparison purpose, then increase/decrease in other treatments must have been mentioned

The reasons of bacterial inoculation are also not highlighted in the introduction section. Why only vetch was inoculated? Why not Medicago inoculation?

The manuscript is much more concentrated on yield-related aspects. Why oxalis has been highlighted when the applied treatments provided no significant control

I suggest to divert the focus and title to yield.

Author Response

The title must not contain results from a 3-year field trial

Authors’ response: We have now removed the “results from a 3-year field trial” (lines 4-5)

The abstract is fine. Just add the age of the orchard

Authors’ response: We have added the age of the orchard to the abstract text (line 27-28).

The last 3 lines of introduction section must be added within the sentences

Authors’ response: This has now been done (lines 79-84)

Please add the age of the orchard in MM section and write cultivar names in inverted commas.

Authors’ response: The orchard age has now been added in the methods (line 91 and 92) and we have put cultivar names in inverted commas throughout the articles (lines 4, 91, 99, 408)

The results of Medicago have been skipped in Tables and figures relating to Oxalis density? If it was for comparison purpose, then increase/decrease in other treatments must have been mentioned.

Authors response:

We do describe the Medicago results in a supplementary Table (see Supplementary Table 1)

However, in the manuscript Table 2. we only compare the three vetch-based cover crop treatments. In these 3 cover crop treatments the legume (vetch) established well and Table 3 analyses the effect of (a) using Rhizobium inoculum for vetch or (b) combining vetch with pea and barley.

To compare the non-inoculated vetch with the non-inoculated Medicago treatment we have included an additional ANOVAs Table in the Supplementary Material (Supplementary Table 1). The main effect of legume species (uninoculated vetch versus uninoculated Medicago) is also already described in the results section which refers to Supplementary Table 1 by the following text:

“However, establishment of the native Medicago species resulted in very poor establishment (9 plants/m2) when compared to vetch cover crops established with untreated seed (156 plants/m2) (Supplementary Table 1).” (lines 226-228)

.

The reasons of bacterial inoculation are also not highlighted in the introduction section.

Authors’ reply: We already describe why Rhizobium inoculation was investigated in the Introduction with the following text:

“One approach to increase N-fixation and availability from cover crops may be to apply Rhizobium inoculum to legume seed. For example, application of a commercial Rhizobium inoculum to clover seed was recently shown to further increase N-levels in soil and N-supply to subsequent wheat crops grown after clover leys in the UK [4,7]. However, this approach has not been evaluated for vetch (Vicia sativa), the main legume species used as cover crop in organic olive orchards in the Mediterranean region [8].” (lines 56-61)

Why only vetch was inoculated? Why not Medicago inoculation?

Authors’ reply: The size of the orchard and the need to have plots with similar size trees did not allow us to include more than 4 treatments; we therefore decided to study the effect of inoculation only with vetch, since this is the most widely used legume cover crop species.

Also, at the time of the experiment no commercial Medicago species-specific Rhizobium strains were available that could have been used for inoculation of Medicago polymorpha seed.

The manuscript is much more concentrated on yield-related aspects. Why oxalis has been highlighted when the applied treatments provided no significant control. I suggest to divert the focus and title to yield.

Authors’ reply: our results show that there were significant differences in Oxalis density between (a) the 3 vetch-based cover crop treatments (Table 2) and (b) the cover crops established with non-inoculated vetch and non-inoculated Medicago seed (Supplementary Table 1).

We therefore feel that it is justified to focus on both effects of Oxalis and effects on olive tree performance parameters.
